# Electronics and Detectors for the Stellar Intensity Interferometer of the ASTRI Mini-Array Telescopes

**DOI:** 10.3390/s23249840

**Published:** 2023-12-15

**Authors:** Giovanni Bonanno, Luca Zampieri, Giampiero Naletto, Lorenzo Paoletti, Giuseppe Romeo, Pietro Bruno, Alessandro Grillo, Gianfranco Occhipinti, Maria Cristina Timpanaro, Carmelo Gargano, Michele Fiori, Gabriele Rodeghiero, Giovanni Pareschi, Salvatore Scuderi, Gino Tosti

**Affiliations:** 1INAF-Osservatorio Astrofisico di Catania, Via Santa Sofia, 78, 95123 Catania, Italy; giuseppe.romeo@inaf.it (G.R.); pietro.bruno@inaf.it (P.B.); alessandro.grillo@inaf.it (A.G.); giovanni.occhipinti@inaf.it (G.O.); maria.timpanaro@inaf.it (M.C.T.); 2INAF-Osservatorio Astronomico di Padova, Vicolo dell’Osservatorio, 5, 35122 Padova, Italy; luca.zampieri@inaf.it (L.Z.); giampiero.naletto@unipd.it (G.N.); lorenzo.paoletti@oapd.inaf.it (L.P.); michele.fiori@inaf.it (M.F.); 3Department of Physics and Astronomy “Galileo Galilei”, University of Padova, Via F. Marzolo, 8, 35131 Padova, Italy; 4INAF-Istituto di Astrofisica Spaziale e Fisica Cosmica, Via Ugo La Malfa 153, 90146 Palermo, Italy; carmelo.gargano@inaf.it; 5INAF-Osservatorio di Astrofisica e Scienza dello Spazio di Bologna, Via Gobetti 93/3, 40129 Bologna, Italy; gabriele.rodeghiero@inaf.it; 6INAF-Osservatorio Astronomico di Brera-Merate, Via Brera 28, 20121 Milano, Italy; giovanni.pareschi@inaf.it; 7INAF-Istituto di Astrofisica Spaziale e Fisica Cosmica, Via Alfonso Corti 12, 20133 Milano, Italy; salvatore.scuderi@inaf.it; 8Dipartimento di Fisica e Geologia, Università di Perugia, Via Alessandro Pascoli, 06123 Perugia, Italy; gino.tosti@unipg.it

**Keywords:** intensity interferometry, silicon photomultiplier, front-end electronics

## Abstract

The ASTRI Mini-Array is an international collaboration led by the Italian National Institute for Astrophysics (INAF) that will operate nine telescopes to perform Cherenkov and optical stellar intensity interferometry (SII) observations. At the focal plane of these telescopes, we are planning to install a stellar intensity interferometry instrument. Here we present the selected design, based on Silicon Photomultiplier (SiPM) detectors matching the telescope point spread function together with dedicated front-end electronics.

## 1. Introduction

Basically, SI^3^ is a fast single-photon counting instrument for performing intensity interferometry observations by using the technique based on the measurement of the second-order coherence of light [1].

Using large collecting area telescopes separated by hundreds of meters, the SII technique allows angular resolutions below 100 micro-arcseconds to be achieved and thus revealing details on the surface and of the environment surrounding bright stars on the sky, which typically have angular diameters of 1–10 milli-arcseconds (mas) [2].

SII was pioneered by Robert Hanbury Brown and Richard Q. Twiss between the 1950sand the 1970s [3]. They built the Narrabri Stellar Intensity Interferometer using twin 6.5 m diameter telescopes movable along a circular track at Narrabri, New South Wales, Australia, and performed the first direct astronomical measurements of stellar radii.

Theoretical and numerical simulations of such observations have been made for several types of objects [3,4,5,6].

SI^3^ is able to detect photon arrival times (1 ns) in a narrow optical window (3–8 nm) centered at a wavelength in the range 420–500 nm [7].

The nine ASTRI telescopes will offer up to 36 different baselines for performing simultaneous measurements and can potentially reach a very high angular resolution (below 100 µarcsec) in the optical band.

Air Cherenkov telescopes have traditionally been equipped with photomultiplier (PMT) detectors, a natural choice since the image point spread function in the focal plane is typically some cm in extent, fitting PMT sizes well [8].

However, the ASTRI telescope, thanks to its optical two-mirror Schwarzschild–Couder design, produces a compact Point Spread Function (PSF) and allows the use of Silicon Photomultiplier (SiPM) detectors that are less fragile and more efficient than Photomultiplier Tubes (PMTs) [9,10].

SiPMs are very suitable devices for this kind of application. However, they are “Photon Number Resolving” detectors, meaning that two or three or more simultaneous photons (typical situation of high-level intensity) will be counted as one, and this, of course, will affect the rate linearity (~18 Mcps accepting a pileup of 10%). To reach the expected and challenging rate of 80 Mcps, a 2 × 2 SiPM array and dedicated front-end electronics have been designed and developed.

In this paper, we describe the adopted solutions in terms of detector focal plane and front-end electronics. The conceived architecture of the whole detection system of the SII Instrument (SI^3^) is also presented. Finally, some preliminary results related to the instrument performance in terms of time resolution and maximum allowable count rate to maintain a non-linearity better than 15% are presented. At the end, the drawback of the count-rate non-linearity due to the intrinsic SiPM architecture is demonstrated.

## 2. Stellar Intensity Interferometry Instrument—SI^3^

In this section, we briefly describe the SI^3^ instrument and the adopted solutions to make intensity interferometry observations feasible with the ASTRI-MA telescopes. A detailed description of SI^3^ can be found in [7].

The observations with the SI^3^ instrument will be performed only during the moon period when the Cherenkov camera is inactive. We use a moveable and motorized arm hosting the optics and the detectors on the telescope focal plane. Instead, during Cherenkov observation, the arm will place the mechanical system in a parking position so as to not affect the observations. Figure 1 depicts the schematic view of ASTRI SI^3^.

The optical design of the SI^3^ module is based on a pre-focal system deployed in front of the telescope Cherenkov focal plane. The optics are essentially a catadioptric system composed of:a 180 mm diameter spherical convex mirrorthree 40–60 mm diameter spherical lensesone or more interference filters to perform multi-narrow-band observations of the continuum and/or of specific spectral lines.

The position of SI^3^ on the ASTRI telescope is shown in Figure 2.

## 3. Front-End Architecture

### 3.1. Focal Plane Detectors

To reach the “design rate” rate of 80 Mcps, we used a 2 × 2 SiPM array as the SI^3^ focal plane detector. Each single SiPM (model S14160-3050HS) is manufactured by Hamamatsu Photonics and has a size of 3 × 3 mm^2^ with a 50 μm cell size. The left panel of Figure 3 shows the schematics of the 2 × 2 detector array, while the right panel depicts the single 3 × 3 mm^2^ Multi Pixel Photon Counter (MPPC) as it appears in the Hamamatsu datasheet.

The technological maturity of this kind of SiPM has reached a very high level in terms of high Photo Detection Efficiency (PDE), low Optical Cross Talk (OCT), and very low afterpulse (about 0.1%) [11,12,13,14,15]. The latter is very important for our application, because if a photoelectron is released after a time longer than the dead time, it may trigger a signal correlated with the real photon event. Since, in intensity interferometry, one searches for correlated signals, such afterpulsing may appear as a false correlation.

In order to have a moderately low Dark Count Rate (DCR), the SiPM detectors will be cooled through a Peltier Thermoelectric Cooler (TEC). A special mechanical housing is designed and capable of hosting the detector PCB and the TEC. This detector housing will be interfaced with the mechanical structure of the focal plane optics.

A key parameter of front-end electronics is the achievable time resolution between pulses. For this reason, in designing the FEE of SI^3^, particular attention was paid to selecting components characterized by high bandwidths so as not to limit the excellent time response offered by SiPM detectors.

### 3.2. MUSIC: An 8-Channel Readout ASIC for SiPM Arrays to Drive the Four Pixels

To optimize the time response and prevent possible degradation of the timing performance, we selected an ASIC chip named Multiple Use SiPM Integrated Circuit (MUSIC) as a SiPM anode readout and preamplifier, designed at the University of Barcelona (ICC-UB), which is capable of driving up to 8 channels (bandwidth: 150 MHz/channel) simultaneously [16]. Figure 4 depicts the block diagram of the various functionalities, while Figure 5 shows a printed circuit board named eMUSIC mini board developed by Scientifica (spin-off of Barcelona University) that adopts the MUSIC chip.

As can be noted from the functional block (Figure 4), the ASIC also has all the circuitry needed to transform an analog signal into a digital signal, i.e., it has a comparator and a programmable threshold generator. Although the chip offers this capability, unfortunately, we cannot use it for the SI^3^, which requires it to work at a threshold that discriminates the single photon, i.e., 0.5–0.7 photon equivalent (p.e.). In fact, the electronic noise of the Mini Board eMUSIC is higher than 1.5 p.e. To avoid waiting for a redesign of the board, we decided to use only the analog part of the chip, and a new 4-channel electronic board has been specifically designed based on opamps. The analog channels achieve excellent SNR, allowing one to clearly identify more than 10 photon peaks in the charge spectrum as depicted in Figure 6 [16].

A very important characteristic of this ASIC chip is the possibility of introducing a derivative circuit via a programmable Pole Zero Cancellation (PZC) in each channel, which allows minimization of the dead time between two consecutive pulses. The PZ cancellation narrows down the SIPM pulse, providing an excellent resolution with an FWHM of about 5 ns when the input SiPM pulse FWHM was longer than 100 ns. Figure 7 illustrates a typical signal before and after PZ cancellation for a 6 × 6 mm^2^ and 50 µm micro-cell SiPM signal. The PZ shaper is employed to reduce the peak duration of the signal and the tail produced by the sensor.

### 3.3. Wideband Low-Noise Amplifiers

As discussed in the previous subsection, we decide to use only the analog part of the ASIC and external amplifiers and comparators to generate the digital signals needed by the Back-End Electronics. For this reason, the output of the eMusic board is amplified by four wideband (0.05–8 GHz) low-noise amplifiers, ZX60-83LN-S+, manufactured Mini-Circuits (see Figure 8). Many examples of SiPM analog front-end electronics can be found in the literature [17,18]. But, to save the work conducted previously with the MUSIC, we added these amplifiers. They use Heterojunction Bipolar Transistor (HBT) technology to deliver ultra-flat gain over a wide frequency range. The high gain, 21 dB, reduces the number of gain stages, lowering the component count and overall system cost.

The purpose of these amplifiers is to increase the signal amplitude at the input of the leading edge discriminator, which works with amplitudes higher than 100 mV pp (see next subsection). In order to minimize the electronic noise due to cabling, we positioned the four amplifiers close to the eMUSIC board and used coaxial cables to connect them to the discriminators. The four amplifiers ZX60 and the Mini Board eMUSIC are placed together into the same box of the detector array board, constituting a first analog stage that we call “PRE-FEE”. This part of the FEE is placed close to the detector to reduce the electro-magnetic noise as much as possible. Figure 9 shows a screenshot of the output of one channel amplifier captured with a 4 GHz bandwidth LeCroy–Teledyne oscilloscope operated in persistence mode.

### 3.4. Leading Edge Discriminator

The part of the FEE that performs the pulse digital conversion is the Leading Edge Discriminator board, placed in a second separate box along with other boards (see the Front-End Electronics box). PRE-FEE and FEE boxes were connected by analog cables for the four signal channels and power cables. As discriminators, we selected the MAX9601, which has two comparators on chip. We preferred not to design a new board, but we used two evaluation boards (EV kit) that mount the MAX 9601 dual channel comparators. The two comparators feature extremely low propagation delay (500 ps) and minimize channel-to-channel skew (10 ps). These features make them ideal for applications where high-fidelity tracking of narrow pulses and low timing dispersion are critical. The outputs are Positive Emitter-Coupled Logic (PECL) complementary digital signals and provide sufficient current to directly drive transmission lines terminated at 50 Ω with PECL output. A MAX 9601 EV board and an example of the output signal with a 100 MHz input signal level of 100 mVp-p are reported in Figure 10.

### 3.5. Threshold Generator Board

To generate the thresholds for the comparators, a dedicated board was designed. It generates four independent thresholds starting from a reference voltage of −1.25 V obtained by a linear low-noise step-down voltage regulator. The setting of the threshold levels is obtained with four DACs (one for each channel) that are set through commands sent via an I^2^C interface by the Control Communication Unit (see next section). The four DACs allow the adjustment (fine-tuning) of the threshold in such a way as to set the same threshold condition for each channel. This procedure of threshold alignment between channels is mandatory because the gain of each SiPM is different from that of the other. Figure 11 shows what the board looks like.

To minimize the analog connection between the DAC’s output and the four thresholds’ inputs of the MAX9601 boards, and to save space, all three boards are mounted in piggyback as can be seen in Section 3.8. 

### 3.6. Control Communication Unit (CCU), Voltage Distribution (VDB) and High Voltage (HV) SIPM Boards

The Control Communication Unit (CCU) and the Voltage Distribution (VDB) boards are placed into the FEE box. Figure 12 shows the block diagram entire FEE. The CCU board hosts an 8-bit microcontroller manufactured by Microchip that acts as the main controller for the digital sections of the FEE and as an analog-to-digital converter for some housekeeping parameters (voltage and current monitoring). The CCU performs the following main tasks:Receiving and executing the commands from the remote control and monitoring system (set the startup configuration and/or the run-time parameters for DAC outputs to the comparator’s thresholds, set the SiPM HV’s parameters); sending to the remote control and monitoring system all the housekeeping data (voltage, current, temperature and so on).Controlling the High-Voltage (HV) SiPM Board circuitry to set the SiPM’s HV.Controlling the Peltier TEC.

The CCU is connected to the remote host via a Universal Asynchronous Receiver- Transmitter (UART) interface followed by a UART to RS485 converter. The CCU drives both the Threshold Generator Board and the HV SiPM Board through two different I^2^C interfaces.

The VDB is actually a set of boards that mainly act as linear voltage regulators for the required power lines; the whole electronics in this assembly need five different voltages and six different distribution lines. The first two boards are a Linear Voltage Regulator Board (LVRB) with six regulators and a power line protection board, named “Crowbars Board” (CB).

The LVRB generates six output voltages starting from two input lines, +12 and −12 V. These two power lines come from a set of DC/DC switching converters that are located in another box to minimize the EM noise. The CB protects the rest of the electronics from excess voltage (i.e., in case of a voltage regulator failure). To minimize the occupied space and the connections, the LVRB, the CB, and the CCU boards are mounted in a piggyback configuration, as described in Section 3.8.

As part of the voltage distribution system, there are two more boards: the HV SiPM Board (HVSB) and the Peltier Thermoelectric Cooler (TEC) board. The HVSB performs the remotely programmable regulation of the SiPM HV through the I^2^C interface from the CCU. This board has an I^2^C ADC that acquires the SiPM temperature and a linear voltage regulator controlled by an I^2^C DAC. The linear voltage regulator is powered by an isolated DC/DC converter from the 24 V line, which outputs 56 V to the linear regulator input.

We created the HV control system with these two stages; as the linear regulators are connected in cascade, they can guarantee a better noise rejection compared to a single switching buck–booster converter. The board has been designed to obtain the SiPM gain compensation as a function of the temperature variation. This feature is very important in SiPM applications where gain stability is mandatory.

The TEC board provides the Voltage/Current desired set point for the TE cooler using the DC voltage coming from the isolated DC/DC converter. This board adopts a linear regulation system instead of the classic, more efficient single-stage switching system, to reduce the output noise and avoid a possible coupling to the SiPM pulse outputs. In fact, the TE cooler is directly coupled to the bottom of the SiPM board.

### 3.7. DC/DC Converters and Switchboard

To power the VDB from the main 24 V DC line and obtain the positive and negative voltages required by the subsequent stages, we used a set of off-the-shelf available DC/DC converters with galvanic isolation. This galvanic isolation is required to reduce the risk of ground loops. Furthermore, to remotely power on/off the whole system and convert the UART TTL level from/to the CCU into a more appropriate RS485 isolated line, we designed a little board named “Switchboard”. This board is based on a solid-state relay that can switch the 24 V on/off and is operated remotely via a set of serial commands by the RS485 line and executed by a dedicated microcontroller placed on the same board. Figure 13 shows the block diagram of the Switchboard and the isolated DC/DC converters.

To reduce the further risk of introducing noise into the FEE, the DC/DC converters and the Switchboard are placed into a third separated box, see next subsection.

### 3.8. Complete Detection System

A block diagram of the entire detection system, including the focal plane FEE made by the eMusic board and the Mini-Circuit amplifiers, is shown in Figure 14. Pictures of all components of the Detection System are shown in Figure 15.

Four blocks can be distinguished:(a)2 × 2 MPPC detector array manufactured by Hamamatsu, with a 3 × 3 mm^2^ area for each quadrant;(b)Front-End Electronics (FEE) made of the eMUSIC MiniBoard, the four Mini-Circuit amplifiers, the two MAX 9601 boards, and the threshold generator board;(c)Voltage Distribution and control boards (VDB);(d)Power supply switch and Control and Communication Unit (CCU).

## 4. Measurement Set-Up

The set-up employed to measure the count rate linearity of all four channels is shown in Figure 16. An integrating sphere is used to uniformly illuminate the detectors and a calibrated photodiode. The count rate of each channel is measured by using an FCA-3000 Tektronix Frequency counter, while the photo-current of the NIST-calibrated photodiode is measured with an ammeter Keithley 6514. We performed several measurements of the MPPC count rate as a function of the current of the photo-diode.

Figure 17 depicts the block diagram of the main functional parts of the detection system. It is possible to identify three blocks that correspond to three boxes. These boxes are those that, at the end, will be mounted on each ASTRI-MA telescope. They are:(a)the PRE-FEE box that contains the 2 × 2 array detector, the eMUSIC board, and the four Mini-Circuit amplifiers,(b)the FEE-VDB-CCU box that contains the discriminator boards, the threshold generator board, the VDB electronics, and part of the CCU(c)the power supply and Switchboard box.

The controls and housekeeping data are provided through a standard RS-485 link. The connections to the back end and the main power supply are also shown.

## 5. Results

The most important parameter to be tested for this application is the linearity over the maximum allowable dynamic range. In other words, the maximum achievable count rate maintaining the ratio between the incident photons and the detected ones is equal to 1. Of course, a 0.5 p.e. threshold has to be selected so as not to lose photoelectrons. SiPMs are intrinsically linear detectors because they are constituted by thousands of Single Photon Avalanche Diodes (SPADs). Each SPAD shows a linear behavior over a very large dynamic range. But in the case of SiPM, the SPADs (cells) are connected in parallel and the output is the sum of each fired cell. Thus, when the number of incident photons becomes larger, the output begins to deviate from the ideal linearity due to the overlapping of pulses. In practice, when the photon flux is sufficiently high to have more photons impinging the detector simultaneously, the SIPM output being the sum of all the fired cells gives only a single pulse with a higher amplitude depending on the number of simultaneous cells firing. This phenomenon is known as the SIPM capability of resolving the photon number. This characteristic is crucial for the instrument linearity: we measure only a pulse in correspondence with more photons because we have to set the threshold to 0.5 p.e., affecting the counting rate.

By using the measurement set up described above, we measured the count rate versus the photo-current generated by the source.

Figure 18 shows the count rate measured by the Tektronix Counter versus the photo-current measured by the Keithley Ammeter for each channel. As expected, when the number of incident photons is low (meaning that the probability of having two simultaneous photons is low), the measured rate is in a linear shape, and instead, the response starts to deviate from linearity above 16–20 Mcps.

We performed repeated measurements aiming at understanding the cause of the non-linearity above 16–20 Mcps. We investigated the response of both the FEE and the SiPM itself, trying to identify the cause of non-linearity. We found that each channel of the FEE is capable of maintaining 100% linearity even above 100 Mcps. As can be noted from Figure 18, we obtained a very good linearity at a rate of 5 Mcps, while the linearity slightly degraded after 10 Mcps. At rate higher than 15 Mcps, we have to consider a 15% deviation from linearity almost acceptable. The deviation from linearity rises to 25% at 18 Mcps.

As mentioned above, in the case of high-intensity photon flux, where the probability of having 2, 3, or more photons within a few nanoseconds is higher, the measured count rate is lower than that expected because our system registers only a single event and cannot count the real number of incident photons.

## 6. Discussion

To better understand the results presented in the previous section, we measured the count rate vs. the photo-current at three different thresholds: one, two, and three p.e. In fact, if it is true that the SiPM counts as one pulse with an amplitude higher than 2, 3, or 4 p.e., then if we measure the count rate by setting a threshold at 2 p.e., we will count the contribution due to the pulses with amplitudes greater than or equal to 2 p.e., which means 2, 3, 4 and so on photons simultaneously. And to account for the rate due to a threshold of 3 or more photons, we repeated the measurements by setting a threshold of 3 p.e.

Considering that for moderately high flux, the probability of having four photos impinging simultaneously on the detector is quite low, we measured the count rate only at these three values of p.e.

At the end, if at the same photocurrent, we add the count rate obtained at 1 p.e., that obtained at 2 p.e., and that obtained at 3 p.e., we, in principle, can recover the count loss due to the above-mentioned effect.

Figure 19 shows the count rate as a function of the photo-current at the 1 p.e. threshold (lower curve), the 2 p.e. threshold (middle curve), and the 3 p.e. threshold (upper curve). As can be clearly seen, at the same photo-current, the count rate rises if we include the contribution of the count rate obtained at the other thresholds. Indeed, including the events at 2 and 3 p.e., the count rate for the same photo-current is more than 30 Mcps for each channel.

This result demonstrates, as expected, that the “photon number resolving” effect typical of SiPMs affects the rate linearity. Thus, for each detector (channel), we have to account for non-linearity at count rate higher than about 20 Mcps due to the SiPM itself and not the FE electronics. This aspect, of course, will have implications for the observation of bright stars.

By splitting the star image in four and using four channels, we can achieve linearity up to ~60 Mcps with deviations of only a few percent and up to 80 Mcps by accepting a 20% deviation from linearity.

## 7. Conclusions

In this paper, we demonstrated that SiPMs can be used as detectors with high time resolution and fast photon counting.

By adopting a 2 × 2 SiPM array as focal plane detector and well-designed front-end electronics, SI^3^ will surely benefit from improved performances. The drawback of a limiting count rate of about 20 Mcps in the linear regime (with a pileup of 15%) for a single pixel can be overcome by sampling the Point Spread Function (PSF) of a star with four quadrants and using a 2 × 2 SiPM array as an SI^3^ focal plane detector. In this case, a total count rate of about 60 Mcps can be achieved without degrading the system linearity.

The use of SiPMs and the challenge of having a very high time resolution led us to adopt solutions in terms of front-end electronics with particular attention on selecting high-bandwidth components so as not to limit the excellent time response offered by SiPM detectors. For this reason, we decided to use an ASIC chip as front-end electronics. This choice was indeed successful, but it imposed a redesign of the architecture.

Finally, some preliminary results related to the instrument performance in terms of time resolution and maximum allowable count rate in the linear regime (to better than 15%) have been presented, and the drawback of the non-linearity caused by the intrinsic SiPM architecture has been discussed. The non-linearity behavior of a single SiPM above ~18 Mcps is overcome by sampling the PSF with a 2 × 2 SiPM array. We also demonstrated that the designed electronics perform very well and are capable of driving the entire detection system above the required performance.

## Figures and Tables

**Figure 1 sensors-23-09840-f001:**
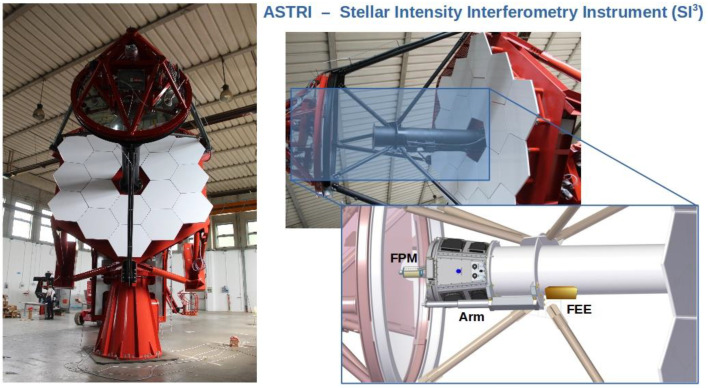
ASTRI telescope during factory assembly (**left**) and schematic view of ASTRI SI^3^ (**right**), with the Focal Plane Module (FPM) deployed in front of the Cherenkov camera by the moveable positioning Arm and the box containing the Front-End Electronics (FEE). The Back-End Electronics (BEE) are located in the telescope cabinet and are not shown.

**Figure 2 sensors-23-09840-f002:**
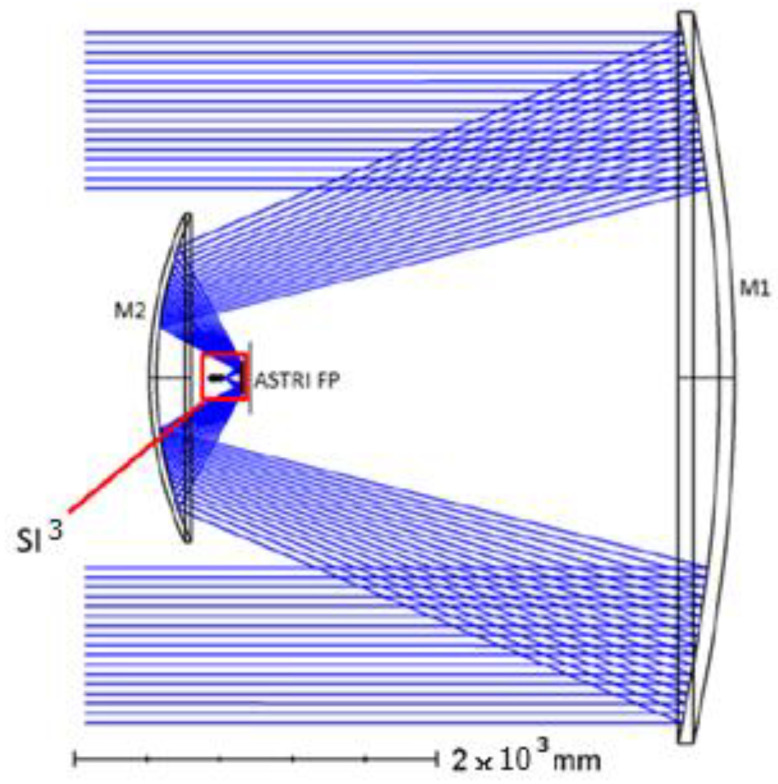
The SI^3^ instrument in its pre-focal position on the ASTRI focal plane.

**Figure 3 sensors-23-09840-f003:**
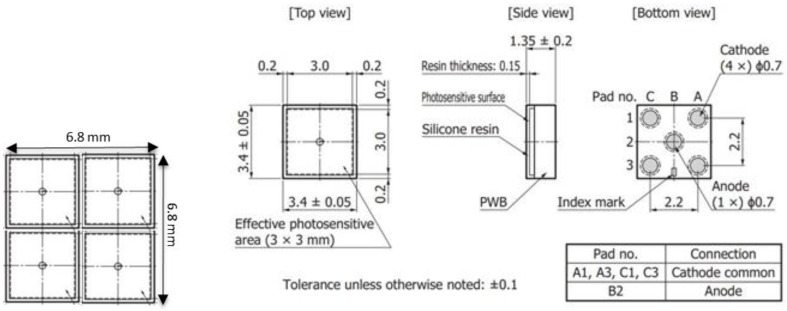
Left panel: schematics of the 2 × 2 detector array. Right panel: single 3 × 3 mm^2^ MPPC from Hamamatsu with a 50 μm cell size.

**Figure 4 sensors-23-09840-f004:**
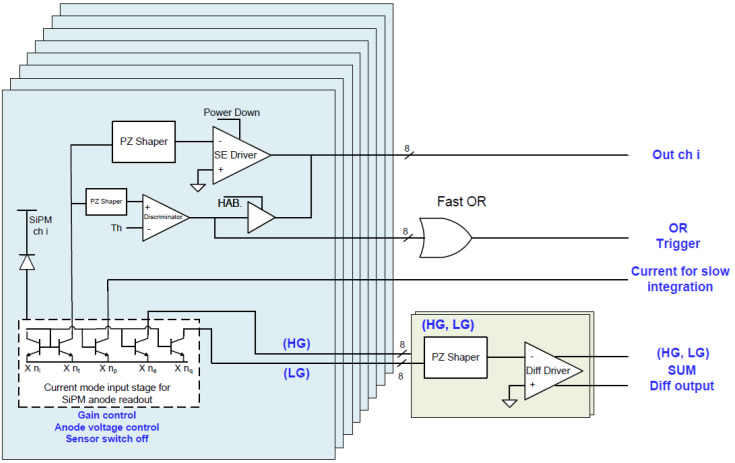
MUSIC functional block diagram.

**Figure 5 sensors-23-09840-f005:**
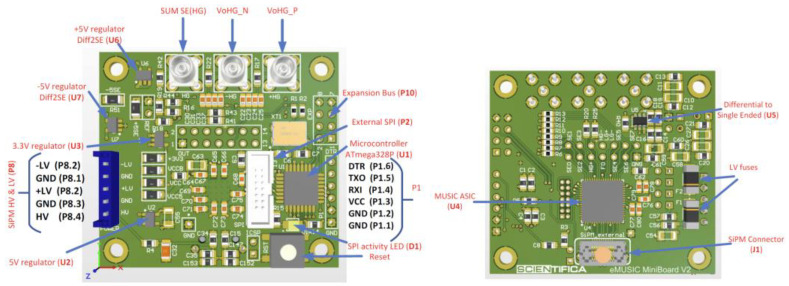
eMUSIC Board Top view (**left**). eMUSIC Board Bottom view where the MUSIC chip is mounted (**right**).

**Figure 6 sensors-23-09840-f006:**
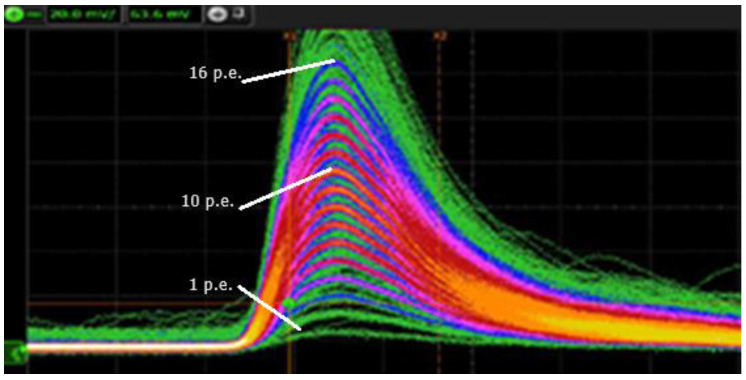
MUSIC analog output for low level illumination. Clear identification of more than 10 photon peaks in the charge spectrum.

**Figure 7 sensors-23-09840-f007:**
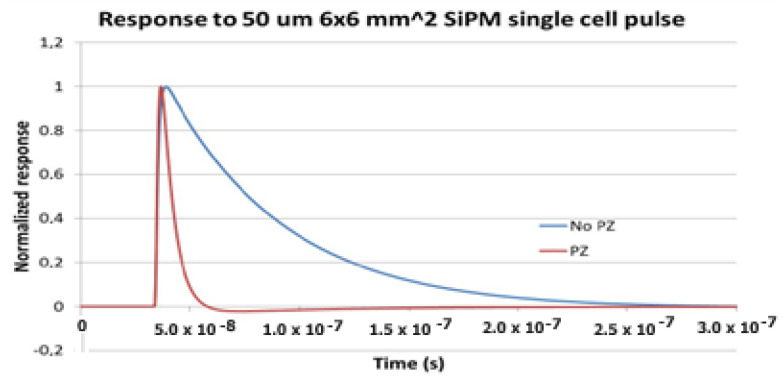
Typical output signal before and after PZ cancellation.

**Figure 8 sensors-23-09840-f008:**
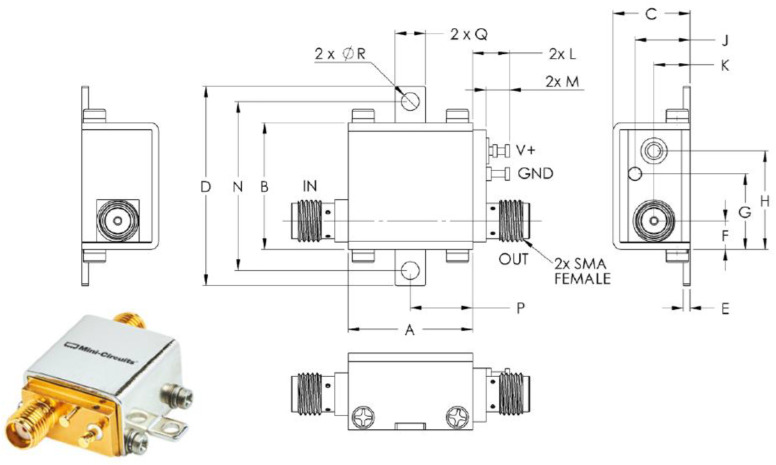
Mini-Circuits ZX60-83LN-S+ low-noise amplifier case style and the outline drawing.

**Figure 9 sensors-23-09840-f009:**
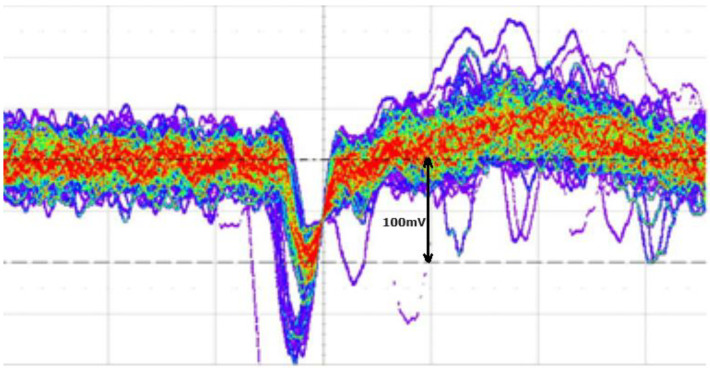
Screenshot of the output of one channel Mini-Circuit amplifier captured with a 4 GHz bandwidth LeCroy–Teledyne oscilloscope in persistence mode. The pulse amplitudes corresponding to 1 p.e. (red plot) and 2 p.e. (violet plot) are clearly visible. The 1 p.e. corresponds to 100 mV.

**Figure 10 sensors-23-09840-f010:**
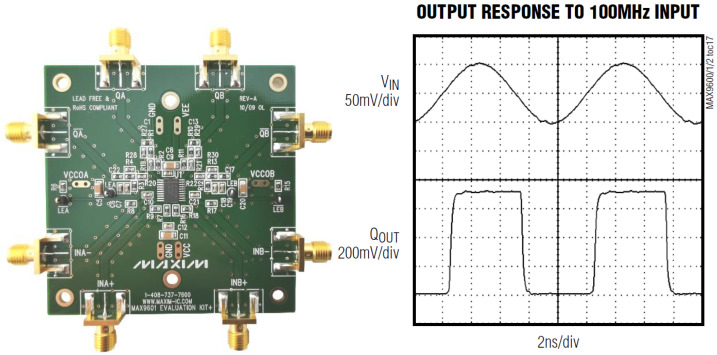
(**Left**): MAX 9601 Evaluation board. (**Right**): example of the output signal with a 100 MHz input signal. Note the input signal level of 100 mV_p-p_.

**Figure 11 sensors-23-09840-f011:**
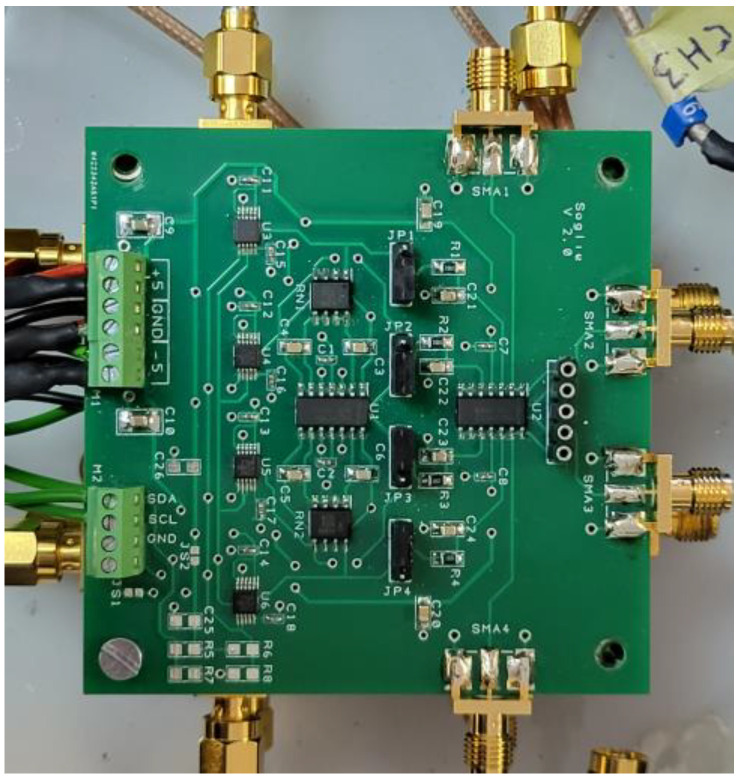
Threshold Generator Board as it appears mounted in the FEE box. The four DACs, recognizable in the left part of the board, are controlled by the Control Communication Unit (CCU).

**Figure 12 sensors-23-09840-f012:**
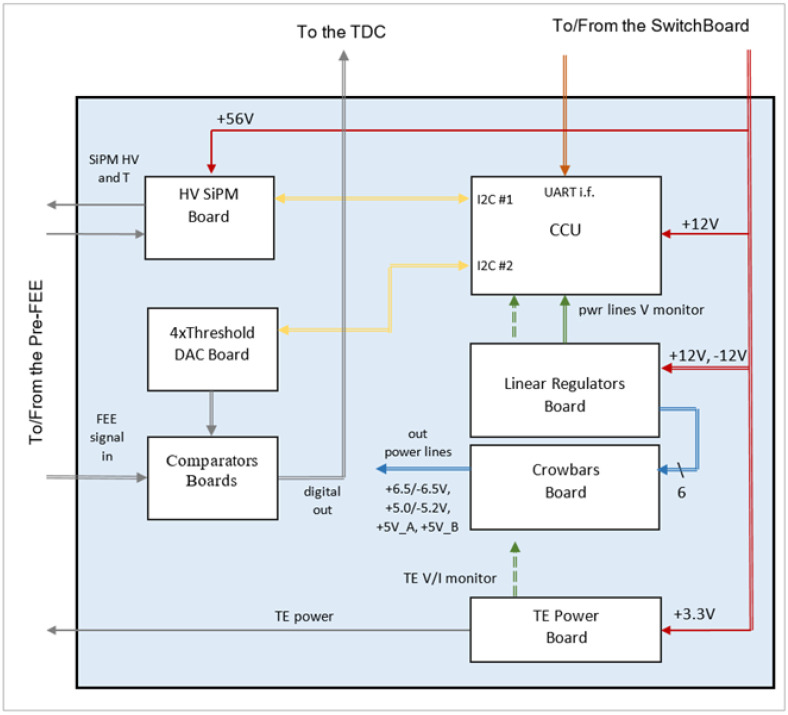
Block diagram of the entire FEE. The LVRB, the CB, and the CCU blocks are shown on the right, while the thresholds and comparator blocks are on the left. The two additional blocks of HV SiPM and TE cooler are also displayed.

**Figure 13 sensors-23-09840-f013:**
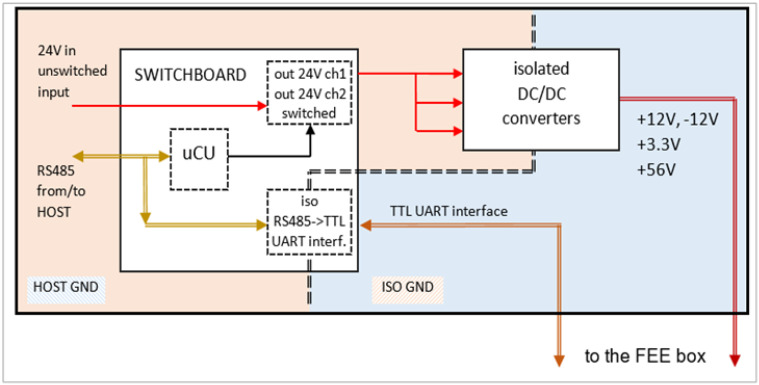
Block diagram of the Switchboard and the DC/DC converters.

**Figure 14 sensors-23-09840-f014:**
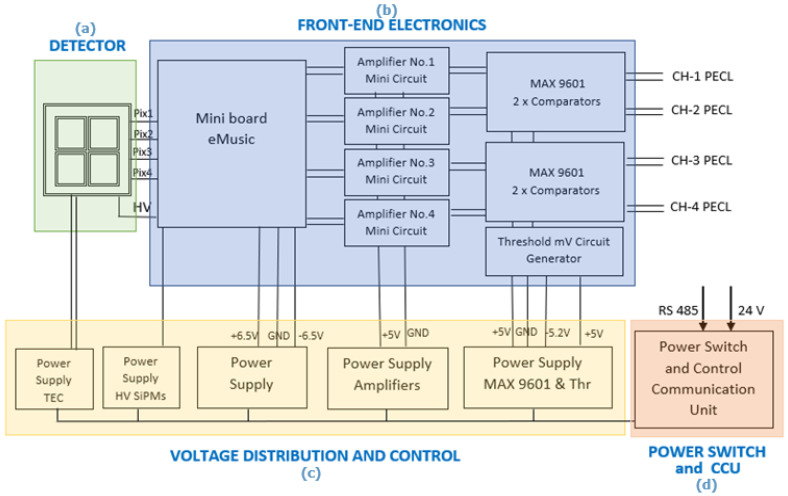
Block diagram of the entire detection system. Four blocks can be identified.

**Figure 15 sensors-23-09840-f015:**
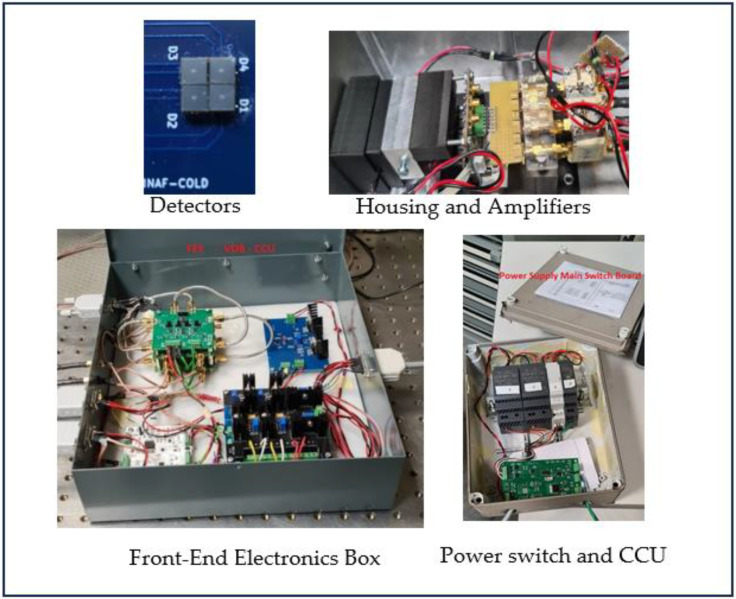
All the constituent parts of the Detection System: the detector board hosting the four MPPCs, the housing with the eMUSIC board and the Mini-Circuits amplifiers, the Front-End Box with the two MAX 9601 boards, the threshold generator, the Voltage Distribution boards, and the power supply Box that also hosts the CCU.

**Figure 16 sensors-23-09840-f016:**
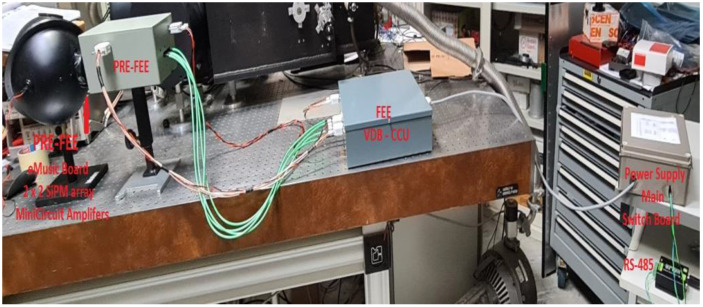
Set-up employed to carry out the count rate linearity measurements of all 4 channels of the detection system, mounted on an optical bench. The two measurement instruments (the counter and the ammeter) do not appear on the image.

**Figure 17 sensors-23-09840-f017:**
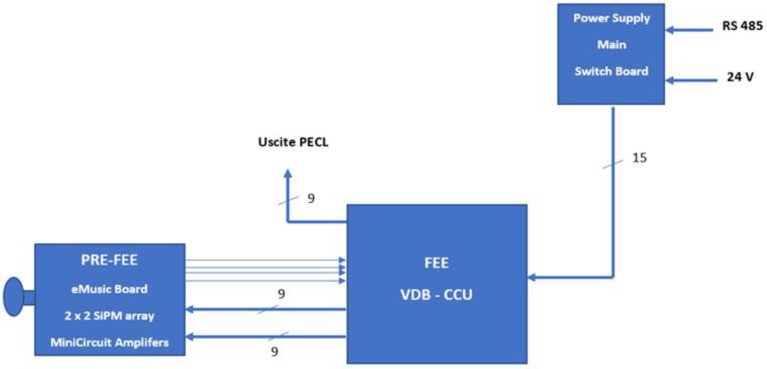
Block diagram of the main functional parts of the detection system. The three blocks correspond to three boxes that, at the end, will be mounted on the ASTRI telescope. The controls and housekeeping data are provided through a standard RS-485 link. The digital output signals are sent directly to the BEE.

**Figure 18 sensors-23-09840-f018:**
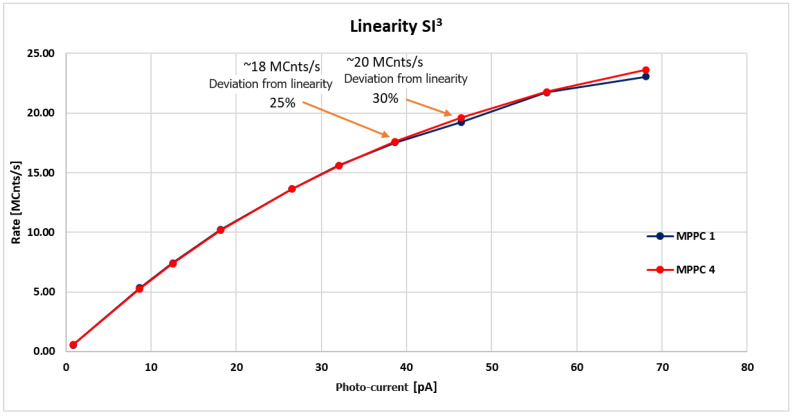
Count rate vs. photo-current for channels 1 and 4. A non-linearity above 18 Mcps is clearly visible.

**Figure 19 sensors-23-09840-f019:**
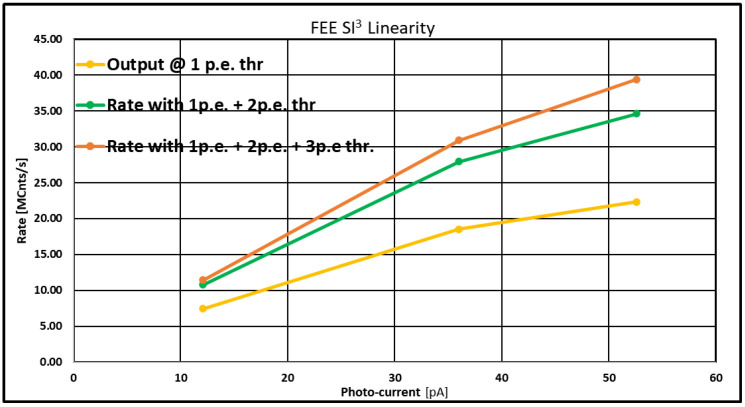
Count rate vs. photo-current. The curves refer to three different cases: 1 p.e. threshold (in yellow), adding the count rate obtained at 1 p.e. and 2 p.e. thresholds (in green), adding the count rate obtained at 1 p.e., 2 p.e., and 3 p.e. thresholds (in orange). As can be seen and as expected, at the same photo-current, the count rate rises if we include the contribution of the other count rates. In this last case, linearity is preserved up to 30 Mcps (instead of 18 Mcps for 1 p.e. threshold case).

## Data Availability

Data are contained within the article.

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
