# Peer review of "Electronics and Detectors for the Stellar Intensity Interferometer of the ASTRI Mini-Array Telescopes"

_sensors, 2023, doi:10.3390/s23249840_

Round 1
Reviewer 1 Report
Comments and Suggestions for Authors
This paper presents a very detailed description of the electronic setup for the intensity interferometer instrumentation on the ASTRI mini-array of Cherenkov telescopes. As such it is a valuable documentation of an unusual type of instrument, planned to be used in the coming years, and quite possibly also inspiring other analogous experiments. The paper is clearly written and should become acceptable for publication following some, mainly cosmetic, amendments.
* The title is rather long and cumbersome; could be shortened to, e.g., “Electronics and detectors for the stellar intensity interferometer of the ASTRI Mini-Array of telescopes”
* Page 1, line 38 – “… determine the second order degree of spatial and temporal coherence …” – No, while the instrument can measure the spatial coherence, the *temporal* coherence cannot realistically be measured. For an optical window of 5 nm, that is on the order of pico- or femtoseconds (the inverse of the frequency bandwidth of light). With great effort, the temporal coherence of starlight can be measured for extremely narrow optical bandpasses (e.g., Tan et al., ApJL 789, L10, 2014) but that is not realistic here, and the reference to temporal coherence should be deleted.
* Numerous abbreviations are introduced. Check that all are explained when first used. E.g., for this paper, the important “MPPC” appears on page 3 but its meaning is explained only on page 11. Also check consistency of designations, e.g., “SI3” appears somewhere for “SI^3”.
* The first paragraphs of Section 3.1 on page 3 are repeated identically to those in the Introduction on page 2. Only keep them in one location.
* Figure 6 is a screenshot that includes also irrelevant border information and as such is not suitable for publication. It should be edited to only retain the curves of significance (with appropriate units marked). Figure 9 is a cleaned screenshot but lacks a clear description of exactly what is shown by its curves of various color.
* Figure 11 is unsharp (at least in the manuscript version sent to me); unless a better version is available, it is not suitable for publishing.
* Figures 13 and 15 are repeated inside Figure 17, and I suggest to remove those, while assuring that the complex Figure 17 is sharp and has a high resolution.
------------
(end)
Comments on the Quality of English LanguageOnly some very minor typos noted, such as "board" >> "boards" on line 199; "Picture the" "Picture of the" on line 269.
Reviewer 2 Report
Comments and Suggestions for Authors
In this paper the authors report the use of a 2x2 array Si PM detectors in the focal plane of a Mini-Telescope array. The advantage of using the SiPM technology is clearly outlined and also demonstrated with the experimental results which confirm the conditions of linear count rate versus the photocurrent. The realized electronic circuits and blocks are well designed for the application. I note the following questions and remarks to consider in the final form of the manuscript :
- I would be helpful to bring details on the technology and performances of the Si-PM detectors.
- In Fig 6 it is not clear which curve correspond to few or 10 photon peaks. Enlarge the numbers on the horizontal scale. Which spectral detection band.
- Which technology and typical performances of the HBT components used for WB low noise amplifiers.
- In fig 9 note the scales on x-y axis.
- Can you develop with more scientific rigor the intrinsic limitations of Si-PM with increasing the source intensity. Estimation of required performances when extending the number of detectors.
To conclude the paper demonstrates significant novel conditions for the count rate detection of single photons in the linear conditions. It can be considered for publication in the journal after revision when taking account of the above comments.
Round 2
Reviewer 2 Report
Comments and Suggestions for Authors
In this revised version the authors have correctly improved and clarified certain points that were requested. Also several new relevant references have been included in the list. I consider that the revised manuscript is now appropriate for publication in the journal.